# Quality Optimization and Evaluation of New Cookie Product with Celery Root Powder Addition

**DOI:** 10.3390/foods13172712

**Published:** 2024-08-27

**Authors:** Milica Nićetin, Jelena Filipović, Ivica Djalović, Dragica Stanković, Goran Trivan, Milenko Košutić, Dragan Živančev, Vladimir Filipović

**Affiliations:** 1Faculty of Technology Novi Sad, University of Novi Sad, Bul. Cara Lazara 1, 21000 Novi Sad, Serbia; vladaf@uns.ac.rs; 2Institute of Food Technology in Novi Sad, University of Novi Sad, Bul. Cara Lazara 1, 21000 Novi Sad, Serbia; jelena.filipovic@fins.uns.ac.rs (J.F.); milenko.kosutic@fins.uns.ac.rs (M.K.); 3Institute of Field and Vegetable Crops, National Institute of the Republic of Serbia, Maksima Gorkog 30, 21000 Novi Sad, Serbia; maizescience@yahoo.com (I.D.); dragan.zivancev@ifvcns.ns.ac.rs (D.Ž.); 4Institute for Multidisciplinary Research, University of Belgrade, Kneza Viseslava 1, 11030 Belgrade, Serbia; dstankovic@imsi.bg.ac.rs (D.S.); goran.trivan@imsi.bg.ac.rs (G.T.)

**Keywords:** combined dehydration, convective drying, lyophilization, *Apium graveolens* L. powder, molasses, biscuit

## Abstract

Combined drying, an energy-efficient method that includes osmotic pretreatment in molasses and shortened successive lyophilization, was used to obtain celery root powder and incorporate it in the formulation of cookies, with the aim of obtaining a new product. Wheat flour was substituted with combinedly dehydrated celery root powder at levels from 0 to 30%, and optimization of the amount of wheat flour substitution regarding technological, sensory and nutritive characteristics was performed. The optimal level of 20% substitution was determined using Z-score analysis, from the aspect of the best nutritive improvement and the mildest adverse impact on the technological and sensory quality. In the second research phase, comparison of the cookies with the 20% celery root powder substitution, dehydrated by different methods, indicated that combined dehydration showed upgraded results in terms of the overall quality of the final product, for 28.85 percentile points higher than cookies with lyophilized and for 65.24 percentile points higher than cookies with the addition of convectively dried celery root powder. The cookie containing celery powder previously osmodehydrated in molasses had higher contents of analyzed minerals (1.2–3.3 times), total phenols (10.8%) and antioxidant activities (14% for DPPH and 4% for ABTS) compared to the cookie with lyophilized powder.

## 1. Introduction

Celery root (*Apium graveolens* L.), a plant of the *Apiaceae* family, contains many compounds beneficial to health, including dietary fibers (primarily cellulose, hemicellulose and pectin), minerals (significant amounts of K, Ca, Mg, P, Zn, Fe), numerous vitamins and essential oils [1,2,3]. Various studies have confirmed the great presence of phenolic compounds and other phytochemicals in celery root (kaempferol, luteolin, apigenin, quercetin, caffeic acid, ferulic acid, p-coumaric acid, tannins, saponins), which are responsible for its pronounced antioxidant, anti-inflammatory and antimicrobial characteristics and role in the prevention of many diseases and disorders [4,5,6,7]. Due to its characteristic organoleptic properties, aromatic smell and pungent taste, celery root is mainly consumed in the daily diet as a vegetable or as a spice in cooking, while its use as an ingredient in food products is limited [8,9].

Although convective drying is one of the most relevant methods for dehydrating vegetables, many studies indicate the disadvantages of this procedure caused by the application of high temperatures, which affect the loss of color, deterioration of texture, change in taste and loss of important nutrients [9,10]. Conversely, the lyophilization method provides great potential for preserving the nutritional and sensory qualities of dried products but at the same time contributes to the increase in process costs and process execution time [11,12]. Using the combined dehydration method presented by the authors Filipović et al. [13], which includes low-energy osmotic pre-treatment in molasses and an abbreviated successive procedure of high-energy-demanding lyophilization, dehydrated raw material enriched with valuable nutrients from molasses is obtained in an economical and energy-enhanced way.

Nowadays, particular focus is on increasing the use value of secondary products of the food industry (such as the above-mentioned sugar beet molasses), which are good sources of natural antioxidants, minerals and other functional ingredients that can be used to enrich food products [14,15]. In addition to the ecological advantage, the implementation of molasses as an osmotic solution in the process of osmotic dehydration is simple; it does not require energy consumption or elevated temperatures. By immersing fresh vegetables in molasses, due to the concentration gradient between the highly concentrated and highly hydrated material, the multicomponent mass transfer results in the loss of water from the submerged vegetable tissue, into which the components from the molasses are simultaneously incorporated [16,17,18].

The consumption of cookies has been widely popularized due to their good shelf life, ready-to-eat form, recognizably pleasant taste and texture and economic availability [19]. The cookies are traditionally produced from wheat flour, with a high percentage of fat and carbohydrates and a small amount of dietary fiber, minerals and other functional components, which can have a negative impact on the health of people who regularly consume them, especially excessively [20,21]. With increasing living standards and health awareness, there is a greater demand for the production of cookies with improved nutritional and functional value [22,23]. The cookie can be a good candidate for the production of upgraded quality food because its nutritional value can be improved by changing its basic composition by flour substitution or supplementation with health-promoting ingredients [22,23,24]. Since any change in the standard dough formulation affects the final sensory and textural properties of the newly formulated cookie, it is necessary to carefully select and optimize the amounts of the ingredients in order to meet consumer expectations [21,25]. 

The goal of the research was structured in two consecutive phases. In the first phase, the goal was set to define the optimal substitution amount of wheat flour with osmodehydrated and lyophilized celery root powder in the cookie, in which the most favorable technological and nutritive profiles were achieved. In the second phase, the influence of different dehydration methods (combined osmodehydration and lyophilization, convective drying and lyophilization) on the same technological and nutritional parameters of cookies with the optimal addition of celery powder was examined.

## 2. Materials and Methods

### 2.1. Material

Fresh celery root (*Apium graveolens* L. var. *rapaceum*, Alabaster variety) was acquired at the local greengrocery in Novi Sad (Serbia), of which the average dry matter content was 9.05%. Sugar beet molasses, used as an osmotic solution in the osmotic pretreatment, was obtained from a sugar factory in Crvenka, Serbia, with an average dry matter content of 86.04%. The following material was used for the preparation of cookies: white wheat flour, type T-400 (moisture content of 14%), produced by “Danubius”, Novi Sad, Serbia; margarine produced by “AD Dijamant”, Zrenjanin, Serbia; sugar produced by “Šajkaška” Žabalj, Serbia; NaCl produced by “SO Produkt”, Stara Pazova, Serbia; and NaHCO_3_, produced by “Aleva”, Novi Kneževac, Serbia.

### 2.2. Combined Method of Dehydration

The combined method of celery root dehydration was carried out in two stages: first the process of osmotic dehydration in molasses, and then lyophilization. Fresh celery root was initially washed with running tap water, dried with paper towels, peeled and cut into cubes of approximately 1 cm × 1 cm × 1 cm. Then the celery cubes were immersed in vessels filled with enough molasses to obtain dehydrating material in an osmotic solution ratio of 1:5 in an effort to prevent excessive dilution of the molasses and slowing down the kinetics of the process. The osmotic dehydration process took place for 5 h at atmospheric pressure in a thermostatic chamber (Memmert IN160, Schwabach, Germany) where the temperature was set and maintained at 20 °C. After 5 h, the osmotically treated celery samples were separated from the molasses, washed with running water to remove excess solution on the surface of the cubes and then blotted with paper towels to remove excessive water. Obtained osmotically dehydrated celery samples were frozen and stored at −30 °C for 24 h and then subjected to the lyophilization process, using the following device: Christ ALPHA1-2 LDPLUS (Osterode am Harz, Germany). The lyophilization parameters were set at the following: pressure of 1.6 Pa, condenser temperature of −57 °C, shelf temperature at room temperature and a process duration of 24 h. After lyophilization, dehydrated samples were finely ground into a powder of uniform particle size using a universal laboratory mill, type WZ-1 (Solem, ZBPP, Bydgszcz, Poland).

### 2.3. Convective Drying

Diced celery samples were dried to a constant mass in a dryer (Instrumentaria, Zagreb, Croatia) at 50 °C and then pulverized using a universal laboratory mill, type WZ-1 (Solem, ZBPP, Bydgoszcz, Poland), to obtain convectively dried celery root powder.

### 2.4. Lyophilization

Fresh samples of diced celery were previously frozen for at least 24 h and then were placed on metal trays of a freeze-dryer (Christ ALPHA1-2 LDPLUS, Osterode am Harz, Germany). The lyophilization process was set to the same parameters as in Section 2.2, with the difference being that the time required to obtain the dried samples was 48 h. After this time, the samples were ground using a universal laboratory mill, type WZ-1 (Solem, ZBPP, Bydgoszcz, Poland).

### 2.5. Cookie Samples Preparation

The design of an experimental plan was based on different cookie dough formulations, which included the standard (sample 1) and modified formulations. The first phase of research defined the first seven cookie sample formulations, where osmodehydrated and lyophilized celery root powder substituted for different quantities of wheat white flour (from 5% wheat white flour substitution with celery powder in sample 2 to 30% in sample 7). After performing the cookies’ quality optimization and defining the optimal level of wheat white flour substitution, in the second phase of the research, two more samples (8 and 9) were produced, where the same level of substitution was done but with different types of celery root dehydration methods (convective drying, sample 8; lyophilization, sample 9).

In Table 1, the formulations of all produced samples in both phases of the research are presented.

The cookie preparation included dough production operations such as mixing, processing and baking in a pilot plant for bakery products of the Institute for Food Technology in Novi Sad, Serbia, in accordance with the AACC method 10–50 D [26], as described by Šobot et al. [17].

### 2.6. Technological Parameters Analysis

The technological quality parameters of the cookies were determined by the AACC 10–50D [26] method. Baking weight loss (BWL) was determined by measuring the weight of cookies before and after the baking stage using Equation (1):(1)$$\mathrm{BWL}\;(\%)=\frac{\;\;m_{0}-m_{t}}{m_{0}}\cdot 100$$
where, *m*_0_ is the cookie’s weight before baking (g) and *m_t_* is the cookie’s weight after baking (g). Analysis was performed on six samples.

Drying weight loss (DWL) was determined by measuring the weight of cookies after the baking stage and after 30 min of cooling at the room temperature, using Equation (2):(2)$$\mathrm{DWL}\;(\%)=\frac{\;m_{0^{\prime }}-m_{t^{\prime }}}{m_{0^{\prime }}}\cdot 100$$
where, *m*_0′_ is the cookie’s weight after baking (g) and *m_t_*_′_ is the cookie’s weight after 30 min of cooling (g). Analysis was performed on six samples.

Measurements of cookies’ dimensions were performed after a 30 min cooling period. Measurements were made of the cookie diameter in the lamination direction (length–L), the cookie diameter perpendicular to the lamination direction (width—W) and cookie thickness (T). The average cookie diameter (R) was determined by the lowest W and the highest L. T was measured by stacking six cookies and measuring their total height. After the first measurement, the six cookies were rearranged into a column, and their height was measured again. Finally, the mean of these measurements was divided by the number of cookies (6) to calculate the mean value of the cookies (T). The expansion factor (R/T) was determined by the ratio of the mean values of R and T, which indicates the deformation of the cookie shape during baking.

### 2.7. Texture Instrumental Analysis

Cookie texture parameters were determined using the texture analyzer TA-XT2 Texture Analyser (Stable Micro Systems, Godalming, UK) equipped with a 25 kg load cell and Knife Edge with Shotted Insert HPD/bs tools. Measurements were performed by applying compression mode at the crosshead speed of 1 mm/s prior, 3 mm/s during and 10 mm/s after the analysis. Using Exponent Stable MicroSystems computer software, version 6.0, maximum force (n) and distance at break were recorded as a function of time and are indicators of cookie hardness. The measurement of the textural parameters of the cookies was performed in six repetitions, in each batch of cookies, 24 h after baking, at a temperature of 25 °C and dimensions of 50 mm × 50 mm.

### 2.8. Color Instrumental Analysis

Color parameters of the cookies’ surfaces were determined in six replications, 24 h after baking, using a Chroma meter (CR-400, Konica, Minolta, Tokyo, Japan) tri-stimulus colorimeter (contact surface diameter: 8 mm). Before measuring the samples, calibration was done using a white color standard. The results of the color analysis are presented according to the CIElab color system, where the coordinates are defined as follows: L—brightness (from 0 (black) to 100 (white)), a—greenness/redness (from −a (green) to +a (red)) and b—blueness/yellowness (from −b (blue) to +b (yellow)) [17,27].

The color variation between the control sample and samples with the celery powder addition (ΔE) was determined by Equation (3):(3)$$\Delta E=\sqrt{{\Delta L^{2}+\Delta a}^{2}+{\Delta b}^{2}}$$
where Δ*L* is the difference in the L parameter between the control and the cookie sample with the celery powder addition, Δ*a* is the difference in the a parameter between the control and the cookie sample with the celery powder addition and Δ*b* is the difference in the b parameter between the control and the cookie sample with the celery powder addition. The measurement of the instrumental color parameters of the cookies was performed in three repetitions.

### 2.9. Descriptive Sensory Analysis

In an effort to obtain a sensory profile of the cookie samples, a panel of ten evaluators with previous experience in evaluating various food products was formed according to the appropriate standard: ISO 6658:2017 [28]. The selection of descriptors for the sensory profiling of the cookie samples using the descriptive sensory method was previously performed by the leading evaluator and further adjusted by the rest of the evaluators to better define the sensory profiles of the cookies. The final list comprised six descriptors, where two descriptors characterized cookie appearance (color intensity and surface appearance), one descriptor characterized deviation from the standard taste, one descriptor characterized deviation from the standard odor and two descriptors were used for the textural properties definition (sensory hardness and fracturability). A seven-point scale described the intensity of each descriptor, where 1 was marked as the lowest intensity and 7 as the highest intensity [29], except for descriptors for taste and odor where optimal descriptor values were set to value 4 and deviations from this value were characterized by different cookie sample sensory attributes. 

Sensory testing of each cookie sample was performed 24 h after baking in the sensory analysis laboratory of the Institute of Food Technology, Novi Sad, Serbia, and designed according to the ISO 8589:2007 standard [30]. The tasting began after panelists filled out the informed consent form following the Declaration of Helsinki guidelines. Cookie samples were served to the panelists on white plastic plates, coded by random three-digit codes from the table of random numbers. The evaluators were given enough water to rinse their mouths between each tasting.

### 2.10. Chemical Analysis

An approximation of the chemical composition of cookie samples was conducted according to AOAC standard methods [31]: protein content (method No. 950.36), starch content (method No. 996.11), total sugars content (method No. 2020.07), cellulose content (method No. 973.18), lipid content (method No. 935.38) and ash content (method No. 930.22). Each measurement was performed in three replications.

### 2.11. Minerals Analysis

The mineral contents of potassium (K), calcium (Ca), magnesium (Mg), iron (Fe), zinc (Zn) and copper (Cu) of the cookies were determined in accordance with the standard methods of AOAC [31]. Minerals were determined by atomic absorption spectrophotometry (method No. 984.27) on a Varian Spectra AA 10 (Varian Techtron Pty Ltd., Mulgvare, Melbourne, VIC, Australia). Each measurement was performed in three repetitions.

### 2.12. Phenol Content and Antioxidant Activity Analysis

For the determination of total phenol content and antioxidant activity, 2.5 mL ethanol, acetic acid and water (50:8:42) were mixed with 500 mg of powdered cookie sample on Vortex for 2 min, then centrifuged at 12,000 rpm for 3 min. The liquid part was collected and filtered through a 0.45 μm filter.

Total phenol contents in cookie extracts were determined by the Folin-Ciocalteau method adapted to microscale [32]. Results were expressed as mg gallic acid equivalents per 100 g cookie (mg GAE/100 g dry matter). The antioxidant activity was determined using two different methods of free radical scavenging: 2,2-diphenyl-1-picrylhydrazyl (DPPH) as described by Tumbas Šaponjac et al. [33] and 2,2′-azino-bis-3-ethylbenzothiazoline-6-sulphonic acid (ABTS) according to Aborus et al. [34]. Results were expressed as μmol of Trolox equivalents per 100 g cookie (μmol TE/100 g dry matter). Phenol content and antioxidant activity testing was performed in three repetitions.

### 2.13. Methods of Statistical Analysis

#### 2.13.1. Analysis of Variance

Analysis of variance (ANOVA) was applied in order to determine the variations’ statistical significance on the set of all cookie samples’ tested quality responses. ANOVA analysis was performed by using STATISTICA 12.0 software (2013) (StatSoft Europe, Hamburg, Germany).

#### 2.13.2. Z-Score Analysis

In the Z-score analysis, min-max normalization is used for cookie samples’ different response values. They are recalculated and presented in a new dimensionless unit system, with the effort of comparisons and further mathematical calculations of different cookie samples’ quality responses [7].

The maximum obtained value of total Z-score values indicates the optimum value of all segment Z-scores mathematically combined in a defined manner, pointing at the optimal combination of all tested quality parameters of cookie samples.

The calculation of individual segment Z-scores is as follows:

Cookie samples’ technological quality segment Z-score:(4)$$S_{1i}=\frac{\sum_{k=1}^{4}\left(1-\frac{x_{ki}-x_{kmin}}{x_{kmax}-x_{kmin}}\right)+\left(\frac{x_{ji}-x_{jmin}}{x_{jmax}-x_{jmin}}\right)}{5}$$
where *x_k_* are BWL, DWL, R and R/T, and *x_j_* is T.

Cookie samples’ texture instrumental analysis segment Z-score:(5)$$S_{2i}=\frac{\sum_{l=1}^{2}\left(1-\frac{x_{li}-x_{lmin}}{x_{lmax}-x_{lmin}}\right)}{2}$$
where *x_l_* are hardness and fracturability.

Cookie samples’ color instrumental analysis segment score:(6)$$S_{3i}=\frac{\sum_{m=1}^{3}\left(\frac{x_{mi}-x_{mmin}}{x_{mmax}-x_{mmin}}\right)+\left(1-\frac{x_{ni}-x_{nmin}}{x_{nmax}-x_{nmin}}\right)}{4}$$
where *x_m_* are L, a and b, and *x_n_* is ΔE.

Cookie samples’ descriptive sensory analysis segment score:(7)$$S_{4i}=\frac{\sum_{n=1}^{3}\left(1-\frac{x_{oi}-x_{omin}}{x_{omax}-x_{omin}}\right)+\left(\frac{x_{pi}-x_{pmin}}{x_{pmax}-x_{pmin}}\right)}{6}$$
where *x_o_* are color intensity, hardness and fracturability, and *x_p_* are surface appearance, taste and odor.

Cookie samples’ chemical composition segment Z-score:(8)$$S_{5i}=\frac{\sum_{r=1}^{5}\left(\frac{x_{qi}-x_{qmin}}{x_{qmax}-x_{qmin}}\right)+\left(1-\frac{x_{ri}-x_{rmin}}{x_{rmax}-x_{rmin}}\right)}{6}$$
where *x_q_* are proteins, starch, total sugar, cellulose and ash, and *x_r_* is lipids.

Cookie samples’ mineral matter content segment Z-score:(9)$$S_{6i}=\frac{+\sum_{s=1}^{6}\left(\frac{x_{si}-x_{smin}}{x_{smax}-x_{smin}}\right)}{6}$$
where *x_s_* are K, Mg, Ca, Fe, Zn and Cu.

Cookie samples’ phenol content and antioxidative activity segment Z-score:(10)$$S_{7i}=\frac{\sum_{t=1}^{3}\left(\frac{x_{ui}-x_{umin}}{x_{umax}-x_{umin}}\right)}{3}$$
where *x_u_* are total phenolic content, DPPH and ABTS.

Total quality cookie samples’ Z-score:(11)$$S_{i}=0.15\cdot {S_{1}}_{i}+0.05\cdot {S_{2}}_{i}+0.1\cdot {S_{3}}_{i}+0.20\cdot {S_{4}}_{i}+0.20\cdot {S_{5}}_{i}+0.2\cdot {S_{6}}_{i}+0.1\cdot {S_{7}}_{i}$$
where cookie samples’ technological quality characteristics Z-score values (*S*_1*i*_ to *S*_3*i*_) contribute 30%, descriptive sensory characteristics Z-score values (*S*_4*i*_) contribute 20% and nutritive quality characteristics Z-score values (*S*_5*i*_ to *S*_7*i*_) contribute 50% to the total Z-score, or total quality.


max [*S_i_*] → optimum
(12)


Z-score values were calculated using Microsoft Excel ver. 2016. (Microsoft Corporation, Redmond, WA, USA).

#### 2.13.3. Principle Component Analysis

Principal component analysis (PCA) was applied as a pattern recognition technique for data analysis. XLSTAT Version 2014 5.03 Add-in (Lumivero, Denver, CO, USA) in Microsoft Excel ver. 2016. (Microsoft Corporation, Redmond, WA, USA) software was used for the PCA calculation.

## 3. Results and Discussion

### 3.1. Technological Quality of Cookies

Replacing wheat flour with any material in the basic formulation of cookies can distinctly change their physicochemical properties, primarily parameters that define technological quality, such as cookie geometry and texture profile [24,35]. During mechanical mixing, the raw materials used in the composition of the cookies interact to finally form a unique shape, consistency, appearance, color and taste in the last stage of baking [25,36]. The influence of substituting flour with combined dried celery root powder at the level of 5 to 30% on the parameters chosen to characterize the technological quality of cookies is shown in the Table 2.

Weight reductions during baking and drying are important indicators of technological quality because they indicate the final texture and yield of the finished product [37]. The results from Table 2 revealed that the addition of different quantities of pulverized celery root in cookie formulations exerted statistically significant influence on the baking weight reduction response. By increasing the amount of celery powder substitution (5%, 10%, 15%, 20%, 25%, 30%), there was a corresponding decrease in baking weight loss in samples 2–7 compared with control sample 1 (without combined dehydrated celery addition). A possible explanation for this observation could be found in the fact that the celery powder added to the cookies is a high cellulosic material compared to an adequate amount of wheat flour, scarce in insoluble dietary fiber [8]. Cellulose fibers can absorb a certain amount of water and probably thereby influence the better distribution of moisture in the dough, so during baking, when the temperature rises, water is more strongly bound, and less water is subject to evaporation [35,38]. This observation is consistent with the research of Wang et al. [8], where it was reported that the water absorption of wheat dough increases with the increase in added celery powder, while the gluten protein network is weakening. The drying weight loss parameters followed the same trend of a significant (*p* < 0.05) reduction with a percentage increase in the quantity of added dehydrated celery. The same proposed explanation that is related to a more uniform moisture distribution and a higher content of bound water in the dough can be attributed to the behavior of the drying weight loss in the tested cookie samples (2–7).

Factors that describe cookie dimensions (thickness, average diameter and diameter-to-thickness ratio) are important for defining the influence of additives on its technological characteristics and the quality control of the finished product [35,37]. The thickness of the cookie is a consequence of the formation of the dough structure during baking through setting the balance between the gluten protein coagulation and the expansion of the dough, due to the steam creation and gas release from leavening agents [24,37]. The addition of dehydrated celery root powder led to a destabilization of the dough structure, which resulted in a decrease in the cookie’s thickness (2–7) compared to the control sample (1). Even the smallest addition level of 5% (sample 2) caused a statistically significant decrease in the thickness of the cookie, and an increase in the addition up to 30% led to obtaining even thinner cookies. Similar results were reported by Wang et al. [8] and Lauková et al. [39], where the addition of celery root powder caused the reduction of a specific volume in breads. These authors stated that replacing wheat flour with celery powder disrupts the homogeneity and continuity of the gluten network, resulting in bread with a lower specific volume and more compact gas cells. The average diameter response, which indicates the extent of dough spread during baking [35], showed the reverse trend. After the addition of celery root powder in an amount greater than 10% (samples 4–7), the diameter of the cookie was statistically significantly increased compared to the control. The combination of the same factors during the dough formation and baking that affect thickness also affects the final diameter and diameter-to-thickness ratio of the cookie. The value of the R/T ratio indicates the extent of a cookie’s shape deformation [27,35]. Increasing the quantity of celery powder in the formulation of the samples (2–7) caused linearly increased shape deformation, statistically significant at all addition levels. 

### 3.2. Textural Parameters of Cookies

In Table 3, the results of instrumental texture analyses of the hardness and fracturability, key parameters on the basis of which consumers evaluate the quality of cookies, are presented. Hardness is defined by the force required to cause complete breaking of the cookie, while fracturability is an indicator of the texture’s fragility and its tendency to crumble [25,36]. With a percentage increase in celery powder content (2–7), there was an increase in cookie hardness, ranging from 2091 to 5299 N, for the cookie samples 1 and 7, respectively. The increase in cookie hardness was directly proportional to the increase in the level of incorporated combinedly dehydrated celery root powder, and these values were statistically significantly higher with the samples containing higher levels of substitution compared to the control sample. However, the differences were not statistically significantly pronounced between samples 2 and 3 in comparison to the control cookie. Sample 7, with a 30% addition level of celery root powder, has the highest value of hardness, about 2.5 times more than the control cookie. This outcome is expected and in agreement with the research of several authors: Galla et al. [25] observed increased hardness of cookies with the addition of spinach powder at a 5–15% level; Sowmya et al. [40] proved that basil-incorporated cookies were harder compared to control cookie; Drisya et al. [41] noticed that cookies’ hardness increased with increased addition of dried *Murraya koenigii* powder; Deepali and Roji [42] reported that biscuits with increased concentrations of coriander powder (10–30%) had greater hardness. All of these studies showed that firmness tends to rise with the addition of fiber content in cookies, because polysaccharides mechanically interfere with gluten proteins’ interaction, reducing their ability to form a strong network. In addition, since fiber absorbs water, the amount of water available for hydration and development of the gluten matrix decreases, leading to a dilution of the gluten protein concentration, which results in a harder cookie texture [25,40,43]. Accordingly, the increased hardness in tested samples compared to the control cookie could be explained by substitution of wheat flour with a corresponding quantity of combinedly dehydrated celery root powder—cellulose-rich material. 

It can also be observed that the results for cookie fracturability follow the same trend. The addition of increasing amounts of celery powder from 0 to 30% led to statistically significantly enhanced values of samples’ fracturability. It was noticeable that the fracturability of the cookies increased with the increase in the fiber content by including non-gluten raw materials (celery and molasses) in the cookie’s formulation, which can interfere with the formation of the gluten matrix by the same mechanism as that proposed in the case of the hardness response discussion [40,42]. These results are correlated with the investigation of Parul et al. [44], where the incorporation of high-fiber spirulina and sorghum flours indicated a higher fracturability of biscuits.

### 3.3. Color Parameters of Cookies

Color is considered an essential factor that directly affects the perception and acceptability of the product by consumers [24]. Four instrumental color parameters of cookies fortified with different levels of combinedly dehydrated celery root powder are presented in Table 4. The results revealed that addition of osmotically pre-treated celery root powder statistically significantly reduced the brightness of cookies in all tested samples, as compared to the control cookie. The contribution to darkening was more pronounced as more celery powder was added to the cookie. This outcome could be attributed to the content of molasses in the combinedly dehydrated celery root powder, which, owing to the presence of colored substances, contributed to the darker color of the final products to which it was added. This was confirmed by the research of Šobot et al. [17] and Filipović et al. [27], where the darker color of the cookies was due to the addition of raw materials (wild garlic, peach) that have been subjected to the osmotic dehydration in molasses. The dark color of sugar beet molasses is a consequence of the formation of melanoidins and caramelization products during the production of sucrose, with the intensity of the melanoidins’ color being up to six times more pronounced than other present colored substances [45,46].

The cookies supplemented with celery root powder showed an increase in the share of red tone, proportional to the increase in the level of the supplement, ranging from 6.55 to 9.67, although the differences in cookies’ redness between samples 1–3, 4–5 and 6–7 were not statistically profound. The cookies enriched with increasing levels of celery root powder trended in the opposite direction in terms of yellowness. Proportional augmentation of celery root powder quantity led to a statistically significant decline in yellow tone compared to the control sample. Based on the measured values of color variations, it can be seen that the addition of combinedly dehydrated celery root had a statistically significant effect on the deviation in the color of the tested samples in relation to the color of the control sample. The most noticeable variation in color was recorded in samples 5–7, where the additive level was higher (20, 25 and 30%), while in cookies 2–4, color change was moderately expressed (ΔΕ = 3.64–13.22).

### 3.4. Sensory Profile of Cookies

Sensory assessment plays an important role during the creation of a new product, and it can be used to define the limits within which it is possible to substitute or incorporate the new addition in formulation without significantly impairing products’ quality or consumers’ acceptability [19,43]. The data related to sensory evaluation of tested cookies with dehydrated celery root previously osmotically treated in molasses are presented in Table 5, from which it can be seen that color intensity followed the same trend as instrumental color measurement for the parameter of brightness. With an increase in the quantity of incorporated celery root powder, there was a statistically significant augmentation in the intensity of the color of the cookie, i.e., a darker appearance of the products. The intensity of the color of the control sample (1) was rated with grade 3.5 as a light ruddy (optimal color), and the addition in the amount from 5% to 30% of celery dehydrated by the combined method (samples 2–7) increased the color from slightly dark to extremely dark.

With the increasing levels of addition of combinedly dehydrated celery root powder in cookies, statistically significant deterioration of cookie samples’ surface appearance was observed.

It can be observed that the descriptors for the taste and odor with the increase in lyophilized celery root powder osmotically pre-treated in molasses deviated from the characteristic cookies’ taste and odor, represented by cookie sample 1. In the samples that include the levels of celery root powder additions from 5% to 20% (samples 2–5), statistically insignificant deviation from the standard cookie taste and smell was noted. These findings could be related to the molasses content of the celery powder supplements, which probably masks the pronounced herbaceous and pungent notes derived from the celery.

The addition of 25% and 30% celery root powder to the cookies’ formulations (samples 6 and 7), had a more adverse effect on the overall flavor, introducing statistically significant molasses notes to the cookies’ sensory characteristics. 

The results of sensory hardness and fracturability analysis are highly correlated with results obtained by instrumental textural analysis (Table 3), underlying that subjective experience of cookie consumption meets the instrumental analysis. Along with the increase in the level of addition of combinedly dehydrated celery root powder, the sensory hardness and fracturability of the cookies increased as well. The same ingredients interaction discussion, proposed in Section 3.2. Textural parameters of cookies, can be referred to for the discussion on the results of sensory hardness and fracturability.

### 3.5. Chemical Composition of Cookies

The results obtained from the analysis of the basic chemical composition of cookies prepared with a varied concentration of added combinedly dehydrated celery root powder and the cookie without the addition of celery powder are shown in Table 6.

It was noted that the addition of celery root powder led to a decrease in protein, starch and lipid content in samples 2–7 compared to the control. The content of starch and lipids in cookies proportionally and statistically significantly decreased at all increasing levels of addition, while for a statistically significant decrease in protein content, the amount of addition must be at least 10%. Protein, starch and lipid content in cookies declined from 10.85% to 8.91%, from 47.33% to 33.30% and from 23.07% to 16.15%, respectively, comparing the control cookie sample with the cookie sample with the highest quantity of substitution. Therefore, in sample 7, the most enriched with celery root powder (30%), compared to the sample with the standard formulation, the decrease in the content of these components was the most pronounced: 21.77% for protein content, 42.13% for starch content and 42.84% for lipid content. This can be explained by the fact that celery root and molasses, which were included in the cookie formulation as a substitute for an equivalent amount of wheat flour, are raw materials with low starch and fat contents and with a lower protein content compared to wheat flour. Similar results were reported by Mitrovski et al. [47], where with an increase in the share of beetroot powder in cookies up to 25%, protein content decreased from 9.17% to 8.94%, and fat content decreased from 25.55% to 25.11%.

Conversely, cookies containing osmotically pre-treated celery root powder showed a statistically significant increase in total sugars, cellulose and ash content compared to the control sample. With the increase in the celery root powder share in cookies 2–7, the percentage of these components in the samples was higher, which may be explained by the relatively higher content of total sugars, cellulose and ash in celery root and molasses in comparison to the same quantity of wheat flour. Compared to the control cookie, the increase in total sugars, cellulose and ash content in tested cookies ranged from 1.5 to 4.1 times, from 1.3 to 2.2 times and 1.6 to 4.4 times, respectively, depending on the level of addition (5–30%). Thorat et al. [48] also confirmed that cookies incorporated with lemongrass powder show increased ash, crude fiber and carbohydrate content.

### 3.6. Mineral Composition of Cookies

Replacing wheat flour with lyophilized celery root previously osmodehydrated in molasses had a statistically significant positive effect on the mineral composition of cookies at all analyzed levels of substitution (Table 7). Since wheat flour contains minerals to a much lesser extent than other types of flour [20,23], increasing the proportion of celery powder including molasses, at a substitution level from 0 to 30%, caused increases in potassium content by 6.85 times, magnesium content by 45.27%, calcium content by 2.17 times, iron content by 2.14 times, zinc content by 1.77 times and copper content by 79.17%. Many studies correlate with these results. Somway et al. [40] reported that the incorporation of basil caused 35.21% higher potassium content, 42.47% higher calcium content, 200% higher iron content and 111.11% higher zinc content in cookies. Deepali and Roji [42] confirmed the improved contents of calcium, iron and phosphorous of biscuits fortified with 30% coriander powder. Agrahar-Murugkar [49] proved higher contents of calcium and iron in biscuits after fortification with cumin and moringa powder.

### 3.7. Antioxidant Activity of Cookies

Total phenolic content and antioxidant activity of cookies incorporated with celery root dehydrated by the combined method were determined by two different methods (DPPH and ABTS) and are present in Table 8. As the level of celery powder in the cookie samples increased from 5 to 30%, the values of TPC, DPPH and ABTS radical-scavenging activity statistically significantly (*p* < 0.05) increased. These results indicate that addition of combinedly dehydrated celery root powder could be an effective way to enhance the antioxidant activity of the cookies. The control sample contained 6.42 mg GAE/100 g phenolic content, and supplementation at levels of 5–30% statistically significantly increased this content by 8.17–46.57 times, respectively. The enhancement of cookies’ TPC content can be attributed to the synergistic contribution of celery root and molasses, which are both reported to be rich in phenolic compounds. In addition, the results from both antioxidant activity determinations showed an increase in initial values from 0.6 (sample 1) to 50.84 µmol TE/100 g for DPPH and from 2.43 (sample 1) to 180.61 µmol TE/100 g for the ABTS method, via the maximum substitution level of wheat flour with celery powder. Similar to the presented results, many authors have found that the addition of materials rich in phenols and other bioactive compounds led to an improvement in the antioxidant properties of the final products [42,47,48].

### 3.8. Optimization of Celery Powder Addition

Z-score analysis was applied in an effort to define the optimal quantity of osmodehydrated and lyophilized celery root powder added to the cookies’ formulation from the aspect of all combined quality characteristics. The change in individual quality characteristics with the celery root powder addition can be observed by analyzing segment Z-score values (from S_1_ to S_7_), where it can be seen that maximal Z-score values for technological quality, textural analysis, instrumental color analysis and descriptive sensory analysis (S_1_–S_4_) were obtained for the cookie sample without the addition of celery root powder (Figure 1). Maximal Z-score values for chemical composition, mineral matter and phenol content and antioxidant activity (S_5_–S_7_), however, were obtained for the cookie sample with the addition of the maximal quantity of celery root powder.

Total Z-score values mathematically combine all tested quality characteristics in the following manner: 15%, 5%, 10%, 20%, 20%, 20% and 10% of technological quality, textural analysis, instrumental color analysis, descriptive sensory analysis, chemical composition, mineral matter, phenol content and antioxidant activity contribution to the total cookies’ quality, respectively. Maximal obtained total Z-score value, with the value of 53.13%, marked cookie sample number 5 (20% of osmodehydrated and lyophilized celery root powder substitution for wheat flour in the cookie’s formulation) as the cookie sample with the maximal total quality, hence marking the amount used in this sample as the optimal quantity of celery root powder substitution.

The obtained results of optimization directed the additional research direction and defined the second research phase—evaluation of the combined osmodehydration and lyophilization method of dehydrating celery root via comparison to the more conventional dehydration methods of convective dehydration and lyophilization and their application to the cookies’ formulation. For the evaluation of wheat flour substitution with the celery root powder obtained by the combined osmodehydration and lyophilization method on the cookies’ quality characteristics, the amount of 20% of wheat flour substitution with the celery root powder was chosen, and cookies with the same amount of wheat flour substitution but with differently dehydrated celery root powder were compared.

### 3.9. The Influence of Dehydration Method of Celery on Technological and Nutritional Quality of Cookies

The addition of the same amount of celery powder (20%) obtained by different dehydration methods resulted in higher values for baking and drying weight loss in samples 8 and 9 (celery root dehydrated by convective and lyophilization methods) compared to sample 5 (dehydrated by the combined method), although these values are not statistically significantly different (Table 9). The lowest BWL and DWL parameters in sample 5 could be elucidated by the fact that the added celery powder in this cookie formulation also contains a certain amount of molasses, which penetrated into the celery tissue via secondary mass transfer during osmotic pre-treatment, and since it has known humectant properties, it can thereby affect water retention in the final product [45,50]. Results from Table 9 showed that with the same addition level of celery root powder in samples 5, 8 and 9, there were no statistically significant differences in the values of the thickness and diameter, indicating that the celery root dehydration method did not have a significant effect on the cookies’ dimensions. Slightly higher thickness and lower diameter, however, were observed in sample 5, with celery powder produced by the combined dehydration method. The influence of the dehydration method shows that the cookie sample with celery dehydrated by the convective drying (8) had a statistically insignificantly higher shape deformation compared to cookies with celery dehydrated by the combined method and lyophilization. Hence, the differences in the values of the R/T ratio of samples 5, 8 and 9 are not expressed to a great extent and do not deviate much from the value for the control sample (1), which indicates that the celery root is well-dried and pulverized at the level of flour particles, and it is well incorporated into the dough structure. Considering that it is preferable that the values of BWL, DWL, R and R/T are lower, while T is higher, it is worth noting that the addition of celery root dehydrated by a combined method, including osmotic pre-treatment in molasses, had a more positive effect on the cookies’ technological quality than the addition of celery root dehydrated by convective and lyophilization procedures.

From Table 9, it can be seen that samples 8 and 9 have higher hardness values compared to sample 5, although the differences were not statistically significant. These results indicated that osmotically pre-treated and successively lyophilized celery root affected the softer texture of cookies compared to celery root dehydrated by lyophilization and convective methods. Molasses, which is used as a medium for osmotic treatment during this process, diffused into the celery tissue, enriching it with its content. Although the same amount of powder is incorporated in samples 5, 8 and 9, in sample 5, the cellulose content is lower, considering that molasses does not contain cellulose. There is a possible explanation for the fact that the sample with celery root dehydrated by the combined method showed the lowest hardness, as it was previously mentioned that the higher fiber content leads to higher hardness and fracturability of cookies. The values showing fracturability of cookies did not differ statistically significantly depending on the type of celery drying. Cookie sample 8, containing celery powder obtained by convective drying, showcased the highest hardness and fracturability values.

When comparing cookies with 20% celery root powder dehydrated by different methods, the brightness parameter of the sample was not statistically significantly different for samples with celery dried by combined and convective methods (samples 5 and 8). It was even shown that the cookie with convectively dried celery was darker than the cookie with combined dried celery, where the darker coloring partly comes from molasses. This outcome is consistent with the findings of Galla et al. [25], where darkness increased with increased content of convectively dried spinach powder content in cookies, due to the browning of spinach carbohydrates during baking. On the other hand, the cookie containing lyophilized celery root powder showed an increase in the brightness parameter. Sample 9 was even lighter than the control sample (Table 4), which indicated that this dehydration technique is successful in maintaining the color of the celery root and affects in a positive manner the final product color. The share of red tone (a values) did not differ significantly between cookie samples fortified with combined and convectively dehydrated celery root, but the a values of both samples were increased when compared to the control cookie. However, the cookie with convectively dehydrated celery root powder has a slightly more pronounced redness. Compared to samples 5 and 8, sample 9 showed a statistically significantly lower proportion of red tone, a somewhat lower value compared to the control sample. In terms of yellow tone, cookies with convectively and combinedly dehydrated celery root showed lower values, while the cookie with lyophilized powder had a higher value, even compared to the cookie without celery (sample 1). Compared to the cookie containing combined dried celery, the cookie containing convectively dried celery exerted higher yellowness. Similar to the presented results, Mitrevski et al. [47] demonstrated that biscuits produced by substituting 15%, 20% and 25% spelt flour with convectively dried beetroot powder showed increases in L and a coordinate values and a decrease in the b coordinate value compared to the control sample without beetroot. The value of the parameter that indicates the variation in color in relation to the cookie with the standard formulation was the highest in sample 5 (ΔΕ = 17.84), slightly lower in sample 8 (ΔΕ = 13.60) and the least expressed in cookie 9 (ΔΕ = 6.63), from where the impact of incorporated molasses in the cookies’ composition on its color change is evident.

Sensory analysis revealed that the color intensity was the highest in the sample enriched with combinedly dehydrated celery root powder, while the sample with convectively dried powder received a similar grade, only slightly lower. The sample with lyophilized powder received the lowest score for color intensity, close to the color intensity of the control sample. These results of subjective cookie evaluations were in accordance with previously reported results of instrumentally detected color parameters. Celery root powder produced by the combined dehydration method caused the higher value of the cookie’s surface appearance descriptor compared to the convectively dried and lyophilized celery root at the same addition level. By observing the influence of the celery drying method on the final taste and smell of biscuits with 20% addition of dehydrated powder, it can be seen that the sample with combinedly dehydrated celery root deviated moderately (grade 3.3 for taste and grade 3.5 for odor) from the optimal descriptors set to value 4. This observation is similar to the findings of the sensory evaluation performed by Pestorić et al. [43], Shuchi et al. [51], Thorat et al. [48], Kajal et al. [52] and Sowmya et al. [40], who all reported that herbal cookies had acceptable sensory attributes. The most pronounced deviation from the standard taste and smell of cookies was characterized in the sample fortified with convectively dried powder (grades 6.5 and 6.3, respectively), followed by the sample enriched with lyophilized celery (grades 6.3 and 6.1, respectively). In sample 8, the burnt notes were more pronounced, and in sample 9, notes originating from celery were more pronounced. These results are consistent with Deepali and Roji [42], who claimed that coriander-fortified biscuits had a lesser score in appearance, aroma, texture, taste and overall acceptability than the control biscuits without any fortification.

The highest sensory hardness and fracturability was shown by the cookie with the addition of convectively dried celery root, the lowest hardness was in the cookie with the combinedly dehydrated celery root and the lowest fracturability was seen in the cookie with lyophilized celery root. These results obtained in sensory evaluation correlated with instrumental analysis of texture, indicating subjectively measurable differences in cookies’ texture depending on the type of celery root powder addition.

The influence of the dehydration technique on celery root, which was used as a substitute for wheat flour in the optimal amount of 20% at the final chemical composition of the cookies, is shown in Table 9. The results indicate that the dehydration method of celery root had no statistically significant effect on the contents of fat and starch in tested cookie samples. On the other hand, the cookie with celery powder previously osmodehydrated in molasses had a higher protein content and higher total sugar and ash contents compared to cookies with celery that was convectively and freeze-dried. It was evident that the higher cookie contents of protein and especially total sugars and ash can be attributed to molasses containing large amounts of sugar (over 50%) and minerals [50]. In terms of cellulose content, it is obvious that samples with the addition of convective and lyophilized celery powder showed higher proportions, because celery is a cellulose-rich material, while molasses does not contain cellulose [53].

The data from Table 9 revealed that the cookie with 20% celery root powder osmotically pre-treated in molasses had statistically significant higher contents of all analyzed mineral matters. The mineral contents of cookies with the same addition of convective and lyophilized celery root powder did not differ statistically significantly. In comparison with the control cookie (Table 7), these samples (8 and 9) showed about 49% higher potassium content, 8.6% higher magnesium content, 39% higher calcium content, 25% higher zinc content and 16% higher copper content, with an exception for iron, where 6% lower content was recorded. The increase in the contents of K, Mg, Ca, Zn and Cu in cookies indicates that celery is a good source of these minerals. In celery root, Krešić et al. [10] reported 308 mg/100 g of potassium and 73 mg/100 g of calcium, while Golubkina et al. [2] demonstrated quantities of 4.7 mg/100 g of iron, 0.5 mg/100 g of copper and 0.7 mg/100 g of zinc. Sample 5, containing celery powder enriched with molasses, gained 383.79% higher potassium content, 30.97% higher magnesium content, 77.76% higher calcium content, 75.8% more iron, 51.28% higher zinc content and 33.33% more copper than the control cookie. The explanation for the greatly improved mineral status can be found in the literature, which confirms that sugar beet molasses is a powerful source of minerals. The dominant macroelement is potassium, which quantitatively accumulates in the molasses during industrial sugar production and varies in amount from 2190 to 6000 mg/100 g, while the dominant microelement is iron, which can vary in the range from 2.7 to 11.7 mg/100 g [16,50]. Mordenti et al. [54] reported that sugar beet molasses contains 70 mg/100 g of calcium, 50 mg/100 g of magnesium, 1.3 mg/100 g of copper and 0.7 mg/100 g of zinc.

Comparison of the samples with the same level of celery root powder incorporation but dehydrated by different methods indicated statistically significantly different TPC, DPPH and ABTS values for all three observed samples. As seen from the data in Table 9, the cookie containing celery powder previously osmotically dehydrated in molasses had 10.8% higher content of total phenols, 14% higher antioxidant activity determined by the DPPH method and 4.01% higher antioxidant activity determined by the ABTS method in relation to the cookie with only lyophilized celery powder. The process of lyophilization has been rated by other authors as relatively successful in preserving antioxidant and phenolic components [11,12], so the higher values of the parameters describing the antioxidant potential of cookie sample 5 most likely come from molasses, which distinguishes the compositions of samples 5 and 9. In the research by Filipčev et al. [50] and Chen et al. [55], sugar beet molasses is characterized as a raw material with excellent antioxidant properties. Conversely, sample 8 prepared by the addition of convectively dried celery root in the formulation had distinctly lower total phenolic content and antioxidant activity as determined by both methods as compared to samples 5 and 9. Sorouret al. [56] analyzed the effect of convective drying at T = 70 °C and T = 90 °C on the concentration of phenols in celery, and significant loss of total phenols was found, especially at higher drying temperatures, which is consistent with the presented results. Salamatullah et al. [57] revealed that after heat treatment, the phenol content in 100 g celeriac decreased from 22.2 mg GAE in the control to 3.0 mg GAE. Ramachandraiah and Chin [58] also proved the loss of celery root phenol content after applying convective drying at T = 50, 70 and 100 °C. Marić et al. [11], in their study, confirmed that lyophilization was a more effective method in retaining the antioxidant activity in carrots compared to convective drying. After drying at T = 50 °C, total phenolic content in carrots was reduced by approximately 50% and after drying at T = 70 °C by approximately 75% in relation to the fresh sample. Kręcisz et al. [9] reported the effect of dehydration methods on the content of bioactive compounds in fresh celery root in the following way: TPC 143.43 mg GAE/100 g d.m.; DPPH 386.01 μmol TE/100 g d.m.; ABTS 721.53 μmol TE/100 g d.m for freeze drying and TPC 128.29 mg GAE/100 g d.m.; DPPH 222.48 μmol TE/100 g d.m.; and ABTS 624.41 μmol TE/100 g d.m for convective drying.

### 3.10. PCA

The ability of PCA to decrease the number of parameters involved in complex systems was used to simplify the correlation structure [59] between 32 responses of technological quality; textural, instrumental color and descriptive sensory analysis; chemical and mineral matter composition; and phenol content and antioxidant activity and all nine cookie samples produced and tested in this research. In an effort for data trends visualization and applied descriptors discriminating efficiency, a scatter plot was made, showcasing the first two principal components of the data matrix. At the x-axis and y-axis, the first and second principal components are placed, respectively, as seen in Figure 2.

From the scatter plot, it can be visually determined that the separation of all tested cookie samples is significant, since they are characterized by different quality responses. By observing the set of cookie samples with the same type of celery root powder addition (samples 1 to 5), it can be seen that the increasing quantity of celery root powder addition affected the transition from negative first principal component values to positive values, without affecting the significant shift in second principal component values, between samples.

Addition of celery root powder dehydrated by convective dehydration and lyophilization to the cookie recipes (samples 8 and 9) caused their location shift to higher second principal component values in comparison to samples with the addition of osmodehydrated and lyophilized celery root powder.

Cookie samples without (sample 1) and with the addition of small quantities of osmodehydrated and lyophilized celery root powder (samples 2 and 3) were characterized by high values of technological characteristics responses; lipid, starch and protein content; surface appearance and lightness. Cookie samples with the addition of more significant quantities of osmodehydrated and lyophilized celery root powder (samples 4–7) were characterized by higher values of all mineral content responses, phenolic content and antioxidant activity, ash and total sugar content and instrumental and descriptive sensory hardness and fracturability.

Cookie samples with the addition of celery root powder dehydrated by convective dehydration and lyophilization were differentiated on the basis of higher values of odor and taste responses.

Analysis of the method quality showed that the first two principal components accounted for 94.66% of the total variance, indicating that the given data description is significant.

All tested responses except taste, odor and cellulose significantly contributed to the first principal component. On the other hand, previously mentioned exceptions significantly contributed to the second principal component.

### 3.11. Evaluation of the Overall Quality of the Cookies with Celery Root Powder Addition

Z-score analysis was applied again, but in this case with the goal of evaluating the quality characteristics of the cookies with the addition of osmodehydrated and lyophilized celery root powder (sample 5) in comparison to the quality of the cookies with the addition of celery root powder dehydrated by more conventional methods of convective dehydration and lyophilization (samples 8 and 9, respectively). 

Analysis of segment Z-score values, in Figure 3, showed that sample 5 was characterized by the highest values for technological quality, textural analysis, chemical composition, mineral matter and phenol content and antioxidant activity, while sample 9 was marked with the highest scores for instrumental color analysis and descriptive sensory analysis. Total Z-score, which mathematically combined segment quality characteristics in the same manner as in the case of the previous Z-score analysis of the quantity of celery root powder addition optimization, showed significant differences between cookies’ quality characteristics. The cookie sample with the addition of osmodehydrated and lyophilized celery root powder had total quality characteristics that were 28.85 percentile points higher than the cookie sample with the addition of lyophilized celery root powder and 65.24 percentile points higher than the cookie sample with the addition of convective celery root powder.

## 4. Conclusions

The results revealed that an increase in the addition of combinedly dehydrated celery root powder in cookies led to decreases in weight reduction during baking and drying; thickness; brightness; yellowness; standard taste and odor; surface appearance; and protein, starch and fat content but increases in average diameter, shape deformation, redness, color variation, color intensity, sensory hardness and fracturability, total sugars, cellulose and ash content, mineral and phenolic content and antioxidant activity as determined by DPPH and ABTS methods. The cookie with the 20% osmodehydrated and lyophilized celery root powder addition with a maximal Z-score value of 53.13% represented the cookie sample with the maximum overall quality and therefore the optimal amount of wheat flour substitution in the cookie formulation.

Considering the influence of the celery root dehydration type on the nutritional and technological quality of cookies with the same substitution level of wheat flour with dried powder, the combined method proved to be more effective than the convective and lyophilization methods. The cookie with celery root powder previously osmodehydrated in molasses had even higher contents of analyzed minerals, protein, total sugar and ash and total phenols and antioxidant activities compared to the cookie with lyophilized powder, although lyophilization is considered one of the best drying methods in terms of preserving nutritional and bioactive compounds. It is worth pointing out that combined drying, which implied pre-treatment in molasses and shortened lyophilization, in addition to affecting the improvement of the final product quality, also has environmental, energy and economic advantages.

Further research should be focused on finalizing the cookie recipe in terms of optimizing other ingredients, besides the wheat flour, as well as testing its acceptability for a wider consumer population and the possibility of market placement.

## Figures and Tables

**Figure 1 foods-13-02712-f001:**
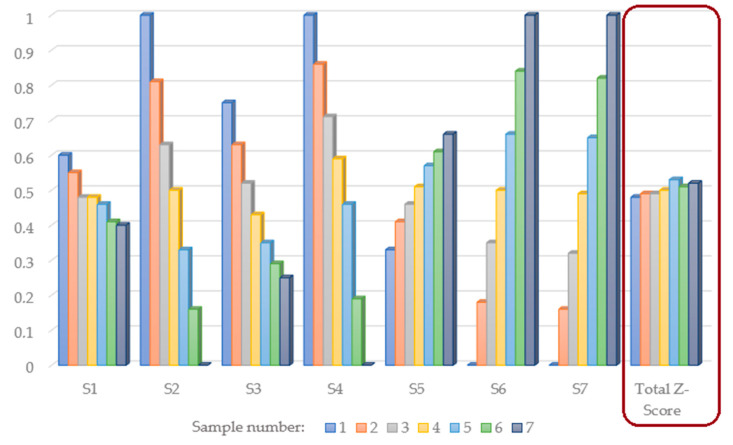
Z-score analysis of cookies with and without the addition of osmodehydrated and lyophilized celery root.

**Figure 2 foods-13-02712-f002:**
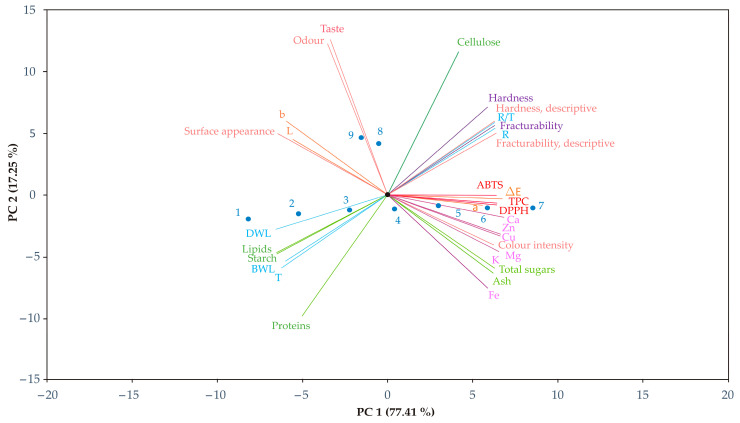
PCA of the tested cookies, with and without the celery root powder substitution.

**Figure 3 foods-13-02712-f003:**
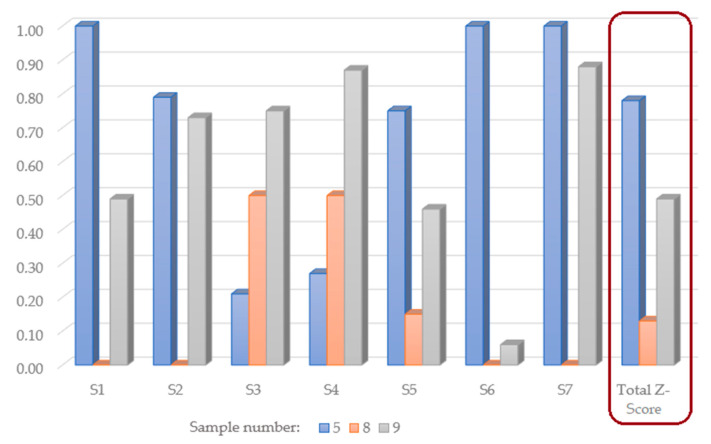
Z-score analysis of cookies with the addition of celery root dehydrated by different treatments.

**Table 1 foods-13-02712-t001:** Experimental design of the flour substitution with dehydrated, pulverized celery root and formulation for tested cookie samples.

Research Phase:	The First							The Second	
Sample no:	1	2	3	4	5	6	7	8	9
Wheat white flour (%)	100	95	90	85	80	75	70	80	80
O.D. + L. celery root (% dry matter (d.m.)) *	0	5	10	15	20	25	30	0	0
C.D. celery root (% d.m.) **	0	0	0	0	0	0	0	20	0
L. celery root (% d.m.) ***	0	0	0	0	0	0	0	0	20
Cookie samples formulation, uniform for all samples									
Wheat white flour/flour mixture (g)	56.25								
Margarine (g)	16.00								
Sugar (g)	0.56								
NaCl (g)	0.53								
NaHCO_3_ (g)	0.63								
Tap water (g)	12.50								

* Osmotically dehydrated and lyophilized, pulverized celery root. ** Convectively dried, pulverized celery root. *** Lyophilized, pulverized celery root.

**Table 2 foods-13-02712-t002:** Cookie samples’ technological quality.

Sample No:	BWL (%)	DWL (%)	T (mm)	R (mm)	R/T
1	25.22 ± 0.51 ^e^	0.96 ±0.13 ^e^	55.05 ± 0.81 ^f^	97.05 ± 0.83 ^a^	1.76 ± 0.01 ^a^
2	21.37 ± 0.73 ^d^	0.91 ± 0.09 ^e^	53.12 ± 0.76 ^e^	98.43 ± 0.49 ^ab^	1.85 ± 0.02 ^b^
3	18.97 ± 1.04 ^c^	0.86 ± 0.06 ^de^	50.43 ± 0.63 ^d^	98.91 ± 0.39 ^a–c^	1.96 ± 0.02 ^c^
4	17.91 ± 0.44 ^bc^	0.77 ± 0.03 ^c–e^	49.74 ± 0.81 ^cd^	99.27 ± 0.23 ^b–d^	2.00 ± 0.03 ^c^
5	17.00 ± 0.46 ^ab^	0.65 ± 0.04 ^bc^	48.42 ± 0.49 ^bc^	99.73 ± 0.38 ^b–d^	2.06 ± 0.01 ^d^
6	16.59 ± 0.36 ^ab^	0.53 ± 0.05 ^ab^	47.12 ± 0.37 ^ab^	100.51 ± 0.73 ^cd^	2.13 ± 0.00 ^e^
7	15.96 ± 0.30 ^a^	0.37 ± 0.01 ^a^	46.24 ± 0.67 ^a^	101.12 ± 1.20 ^d^	2.19 ± 0.01 ^f^

BWL—baking weight loss, DWL—drying weight loss, T—cookie samples’ thickness, R—average diameter, R/T—diameter-to-thickness ratio; results are shown as average value ± standard deviation of six replications. ^a–f^ Different letters in superscript of the same table column indicate the statistically significant difference between values, at a level of significance of *p* < 0.05 (based on post-hoc Tukey HSD test).

**Table 3 foods-13-02712-t003:** Cookie samples’ textural analysis responses.

Sample No:	Hardness (N)	Fracturability (mm)
1	2091.73 ± 193.53 ^a^	1.30 ± 0.20 ^a^
2	2483.43 ± 204.43 ^ab^	1.71 ± 0.12 ^ab^
3	2798.05 ± 424.90 ^ab^	2.09 ± 0.09 ^bc^
4	3297.34 ± 241.17 ^bc^	2.27 ± 0.19 ^cd^
5	3923.50 ± 300.53 ^cd^	2.49 ± 0.16 ^c–e^
6	4643.45 ± 401.27 ^de^	2.65 ± 0.21 ^de^
7	5299.71 ± 534.12 ^e^	2.84 ± 0.25 ^e^

Results are shown as average value ± standard deviation of six replications; ^a–e^ different letters in superscript of the same table column indicate the statistically significant difference between values, at a level of significance of *p* < 0.05 (based on post-hoc Tukey HSD test).

**Table 4 foods-13-02712-t004:** Cookie samples’ instrumental color analysis responses.

Sample No:	L	a	b	ΔΕ
1	57.79 ± 1.01 ^f^	6.55 ± 0.13 ^ab^	22.72 ± 0.29 ^f^	0
2	55.19 ± 1.09 ^e^	6.81 ± 0.19 ^ab^	20.19 ± 0.18 ^e^	3.64 ± 0.02 ^a^
3	50.97 ± 0.51 ^d^	7.35 ± 0.28 ^bc^	18.11 ± 0.13 ^d^	8.27 ± 0.52 ^c^
4	46.53 ± 0.61 ^c^	8.24 ± 0.47 ^cd^	16.01 ± 0.29 ^c^	13.22 ± 0.38 ^d^
5	41.98 ± 0.37 ^b^	8.99 ± 0.57 ^de^	14.83 ± 0.30 ^b^	17.84 ± 0.62 ^e^
6	39.68 ± 0.49 ^ab^	9.37 ± 0.30 ^e^	13.71 ± 0.20 ^a^	20.42 ± 0.52 ^f^
7	37.61 ± 0.78 ^a^	9.67 ± 0.39 ^e^	12.97 ± 0.17 ^a^	22.63 ± 0.29 ^g^

L—brightness, a—greenness/redness, b—blueness/yellowness, ΔΕ—color variation. Results are shown as average value ± standard deviation of six replications. ^a–g^ Different letters in superscript of the same table column indicate the statistically significant difference between values at a level of significance of *p* < 0.05 (based on post-hoc Tukey HSD test).

**Table 5 foods-13-02712-t005:** Cookie samples’ descriptive sensory analysis responses.

Sample No:	Color Intensity	Surface Appearance	Taste	Odor	Hardness	Fractur-Ability
1	3.5 ± 0.2 ^ab^	5.9 ± 0.3 ^e^	4.0 ± 0.0 ^b^	4.0 ± 0.1 ^c^	3.0 ± 0.2 ^a^	4.0 ± 0.0 ^a^
2	4.1 ± 0.3 ^bc^	5.3 ± 0.2 ^de^	3.8 ± 0.2 ^b^	3.9 ± 0.1 ^c^	3.5 ± 0.3 ^ab^	4.6 ± 0.0 ^ab^
3	4.5 ± 0.3 ^cd^	4.5 ± 0.6 ^cd^	3.6 ± 0.4 ^b^	3.7 ± 0.4 ^c^	4.2 ± 0.5 ^a-c^	4.9 ± 0.7 ^ab^
4	5.1 ± 0.4 ^de^	3.9 ± 0.6 ^bc^	3.5 ± 0.1 ^b^	3.6 ± 0.1 ^c^	4.6 ± 0.3 ^bc^	5.5 ± 0.0 ^bc^
5	5.6 ± 0.2 ^ef^	3.5 ± 0.4 ^bc^	3.3 ± 0.5 ^b^	3.5 ± 0.4 ^c^	5.0 ± 0.5 ^cd^	6.1 ± 0.1 ^cd^
6	6.1 ± 0.3 ^fg^	2.9 ± 0.2 ^ab^	2.4 ± 0.2 ^a^	2.6 ± 0.1 ^b^	5.5 ± 0.5 ^cd^	6.9 ± 0.1 ^d^
7	6.6 ± 0.4 ^g^	2.0 ± 0.3 ^a^	2.0 ± 0.2 ^a^	1.9 ± 0.1 ^a^	6.1 ± 0.3 ^d^	7.0 ± 0.0 ^d^

Results are shown as average value ± standard deviation of ten replications (10 assessors). ^a–g^ Different letters in superscript of the same table column indicate the statistically significant difference between values at a level of significance of *p* < 0.05 (based on post-hoc Tukey HSD test).

**Table 6 foods-13-02712-t006:** Cookie samples’ chemical composition.

Sample No:	Proteins (% d.m.)	Starch (% d.m.)	Total Sugars (% d.m.)	Cellulose (% d.m.)	Lipids (% d.m.)	Ash (% d.m.)
1	10.85 ± 0.10 ^e^	47.33 ± 0.27 ^g^	2.17 ± 0.03 ^a^	0.32 ± 0.01 ^a^	23.07 ± 0.19 ^g^	0.41 ± 0.00 ^a^
2	10.65 ± 0.06 ^e^	45.00 ± 0.34 ^f^	3.35 ± 0.01 ^c^	0.41 ± 0.01 ^b^	21.99 ± 0.13 ^f^	0.64 ± 0.01 ^c^
3	10.27 ± 0.17 ^d^	42.64 ± 0.38 ^e^	4.59 ± 0.05 ^d^	0.46 ± 0.01 ^c^	20.70 ± 0.17 ^e^	0.88 ± 0.00 ^d^
4	9.99 ± 0.08 ^d^	40.31 ± 0.43 ^d^	5.70 ± 0.05 ^e^	0.52 ± 0.00 ^d^	19.61 ± 0.09 ^d^	1.11 ± 0.00 ^e^
5	9.67 ± 0.07 ^c^	38.06 ± 0.39 ^c^	6.84 ± 0.04 ^f^	0.59 ± 0.01 ^e^	18.43 ± 0.21 ^c^	1.35 ± 0.01 ^f^
6	9.24 ± 0.13 ^b^	35.71 ± 0.30 ^b^	7.99 ± 0.06 ^g^	0.65 ± 0.00 ^f^	17.20 ± 0.13 ^b^	1.59 ± 0.01 ^g^
7	8.91 ± 0.07 ^a^	33.30 ± 0.27 ^a^	9.09 ± 0.06 ^h^	0.71 ± 0.01 ^g^	16.15 ± 0.09 ^a^	1.82 ± 0.01 ^h^

Results are shown as average value ± standard deviation of six replications. ^a–h^ Different letters in superscript of the same table column indicate the statistically significant difference between values at a level of significance of *p* < 0.05 (based on post-hoc Tukey HSD test).

**Table 7 foods-13-02712-t007:** Cookie samples’ mineral contents.

Sample No:	K (mg/100 g d.m.)	Mg (mg/100 g d.m.)	Ca (mg/100 g d.m.)	Fe (mg/100 g d.m.)	Zn (mg/100 g d.m.)	Cu (mg/100 g d.m.)
1	90.63 ± 0.57 ^a^	16.79 ± 0.15 ^a^	24.54 ± 0.19 ^a^	1.20 ± 0.01 ^b^	0.39 ± 0.00 ^a^	0.24 ± 0.00 ^a^
2	179.50 ± 0.66 ^c^	18.22 ± 0.18 ^b^	29.43 ± 0.20 ^b^	1.42 ± 0.00 ^c^	0.48 ± 0.00 ^b^	0.26 ± 0.00 ^b^
3	265.84 ± 3.19 ^d^	19.55 ± 0.18 ^c^	33.99 ± 0.13 ^c^	1.64 ± 0.01 ^d^	0.53 ± 0.01 ^c^	0.30 ± 0.00 ^d^
4	353.43 ± 4.85 ^e^	20.79 ± 0.16 ^d^	39.00 ± 0.45 ^d^	1.89 ± 0.02 ^e^	0.55 ± 0.00 ^d^	0.33 ± 0.00 ^e^
5	438.46 ± 3.14 ^f^	21.99 ± 0.13 ^e^	43.60 ± 0.26 ^e^	2.11 ± 0.02 ^f^	0.59 ± 0.00 ^e^	0.36 ± 0.00 ^f^
6	530.73 ± 5.80 ^g^	23.30 ± 0.23 ^f^	49.01 ± 0.67 ^f^	2.30 ± 0.01 ^g^	0.67 ± 0.01 ^f^	0.39 ± 0.00 ^g^
7	620.76 ± 5.21 ^h^	24.39 ± 0.06 ^g^	53.20 ± 0.46 ^g^	2.57 ± 0.02 ^h^	0.69 ± 0.01 ^g^	0.43 ± 0.00 ^h^

Results are shown as average value ± standard deviation of six replications. ^a–h^ Different letters in superscript of the same table column indicate the statistically significant difference between values at a level of significance of *p* < 0.05 (based on post-hoc Tukey HSD test).

**Table 8 foods-13-02712-t008:** Cookie samples’ phenol content and antioxidant activity.

Sample No:	Total Phenolic Content (mg GAE/100 g d.m.)	Antioxidant Activity by DPPH (µmol TE/100 g d.m.)	Antioxidant Activity by ABTS (µmol TE/100 g d.m.)
1	6.42 ± 0.04 ^a^	0.60 ± 0.00 ^a^	2.43 ± 0.02 ^a^
2	52.46 ± 0.49 ^b^	8.57 ± 0.04 ^b^	32.44 ± 0.28 ^b^
3	97.82 ± 1.15 ^c^	17.17 ± 0.12 ^c^	60.46 ± 0.29 ^d^
4	141.69 ± 0.72 ^d^	26.10 ± 0.28 ^d^	90.73 ± 0.51 ^e^
5	189.41 ± 2.54 ^f^	33.97 ± 0.12 ^f^	119.71 ± 1.02 ^g^
6	234.76 ± 1.95 ^g^	42.47 ± 0.12 ^g^	153.76 ± 1.63 ^h^
7	299.46 ± 2.02 ^h^	50.84 ± 0.35 ^h^	180.61 ± 1.95 ^i^

Results are shown as average value ± standard deviation of six replications. ^a–i^ Different letters in superscript of the same table column indicate the statistically significant difference between values at a level of significance of *p* < 0.05 (based on post-hoc Tukey HSD test).

**Table 9 foods-13-02712-t009:** Cookie samples’ quality.

	Technological Quality Responses					Textural Analysis Responses			Instrumental Color Analysis Responses			
Sample No:	BWL (%)	DWL (%)	T (mm)	R (mm)	R/T	Hardness (n)		Fracturability (mm)	L	a	B	ΔΕ
5 *	17.00 ± 0.46 ^ab^	0.65 ± 0.04 ^bc^	48.42 ± 0.49 ^bc^	99.73 ± 0.38 ^b–d^	2.06 ± 0.01 ^d^	3923.50 ± 300.53 ^cd^		2.49 ± 0.16 ^c–e^	41.98 ± 0.37 ^b^	8.99 ± 0.57 ^de^	14.83 ± 0.30 ^b^	17.84 ± 0.62 ^e^
8	17.53 ± 0.84 ^a-c^	0.69 ± 0.09 ^b–d^	48.05 ± 0.29 ^bc^	100.21 ± 0.67 ^b–d^	2.09 ± 0.00 ^d^	4713.87 ± 155.54 ^de^		2.57 ± 0.26 ^c–e^	45.06 ± 0.25 ^c^	9.46 ± 0.47 ^e^	18.93 ± 0.33 ^d^	13.60 ± 0.77 ^d^
9	17.18 ± 0.19 ^ab^	0.67 ± 0.04 ^b–d^	48.11 ± 0.41 ^bc^	99.83 ± 0.48 ^b–d^	2.08 ± 0.00 ^d^	4343.45 ± 691.20 ^c–e^	2.43 ± 0.10 ^c–e^	64.34 ± 1.44 ^g^		6.07 ± 0.10 ^a^	23.55 ± 0.72 ^f^	6.63 ± 0.48 ^b^
	**Descriptive Sensory Analysis Responses**						**Chemical Composition Responses**					
**Sample No:**	**Color Intensity**	**Surface Appearance**	**Taste**	**Odor**	**Hardness**	**Fractur-ability**	**Proteins** **(% d.m.)**	**Starch** **(% d.m.)**	**Total Sugars (% d.m.)**	**Cellulose** **(% d.m.)**	**Lipids** **(% d.m.)**	**Ash** **(% d.m.)**
5 *	5.6 ± 0.2 ^ef^	3.5 ± 0.4 ^bc^	3.3 ± 0.5 ^b^	3.5 ± 0.4 ^c^	5.0 ± 0.5 ^cd^	6.1 ± 0.1 ^cd^	9.67 ± 0.07 ^c^	38.06 ± 0.39 ^c^	6.84 ± 0.04 ^f^	0.59 ± 0.01 ^a^	18.43 ± 0.21 ^c^	1.35 ± 0.01 ^f^
8	5.0 ± 0.2 ^de^	5.0 ± 0.0 ^de^	6.5 ± 0.4 ^c^	6.3 ± 0.0 ^d^	5.3 ± 0.8 ^cd^	6.4 ± 0.6 ^cd^	8.84 ± 0.12 ^a^	38.03 ± 0.63 ^c^	2.78 ± 0.02 ^b^	0.79 ± 0.01 ^b^	18.53 ± 0.10 ^c^	0.49 ± 0.01 ^b^
9	3.3 ± 0.3 ^a^	5.6 ± 0.3 ^e^	6.3 ± 0.3 ^c^	6.1 ± 0.2 ^d^	5.2 ± 0.7 ^cd^	5.9 ± 0.2 ^c^	8.95 ± 0.09 ^ab^	38.09 ± 0.42 ^c^	2.81 ± 0.01 ^b^	0.82 ± 0.01 ^c^	18.47 ± 0.06 ^c^	0.48 ± 0.00 ^b^
	**Mineral Content Responses**						**Phenol Content and Antioxidative Activity Responses**					
**Sample** **No:**	**K (mg/100 g d.m.)**	**Mg (mg/100 g d.m.)**	**Ca (mg/100 g d.m.)**	**Fe (mg/100 g d.m.)**	**Zn (mg/100 g d.m.)**	**Cu (mg/100 g d.m.)**	**Total Phenolic Content (mg GAE/100 g d.m.)**		**Antioxidative Activity by DPPH (** **µmol TE/100 g d.m.)**		**Antioxidative Activity by ABTS** **(µmol TE/100 g d.m.)**	
5 *	438.46 ± 3.14 ^f^	21.99 ± 0.13 ^e^	43.60 ± 0.26 ^e^	2.11 ± 0.02 ^f^	0.59 ± 0.00 ^e^	0.36 ± 0.00 ^f^	189.41 ± 2.54 ^f^		33.97 ± 0.12 ^f^		119.71 ± 1.02 ^g^	
8	135.46 ± 0.29 ^b^	18.24 ± 0.11 ^b^	34.01 ± 0.17 ^c^	1.13 ± 0.01 ^a^	0.48 ± 0.00 ^b^	0.27 ± 0.00 ^c^	51.76 ± 0.49 ^b^		8.43 ± 0.06 ^b^		35.70 ± 0.32 ^c^	
9	134.13 ± 1.75 ^b^	18.23 ± 0.09 ^b^	34.15 ± 0.17 ^c^	1.14 ± 0.01 ^a^	0.49 ± 0.01 ^b^	0.29 ± 0.00 ^d^	170.94 ± 0.88 ^e^		29.73 ± 0.22 ^e^		115.09 ± 0.82 ^f^	

* Response values for sample 5 are repeatedly shown in Table 9. for easier comparison with the response values of samples 8 and 9. ^a–g^ Different letters in superscript of the same table column indicate the statistically significant difference between values at a level of significance of *p* < 0.05 (based on post-hoc Tukey HSD test).

## Data Availability

The original contributions presented in the study are included in the article material; further inquiries can be directed to the corresponding author.

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
