# Peer review of "Quality Optimization and Evaluation of New Cookie Product with Celery Root Powder Addition"

_foods, 2024, doi:10.3390/foods13172712_

Round 1

Reviewer 1 Report

Comments and Suggestions for Authors

The aim of this research is to create a new type of cookie by using celery root powder, which is made through an energy-saving method called combined drying (osmotic pretreatment in molasses followed by lyophilization). The goal is to substitute wheat flour with this powder in different amounts (0% to 30%) and find the best substitution level that balances good nutrition, taste, and texture. The study determined that using 20% celery root powder is optimal. In the next phase, the study compared different drying methods for celery root and found that combined drying produced the highest quality cookies compared to other methods. When I look take a look at the aim, the article is interesting, but some issues should be taken into account for acceptation:

Title: please provide different keywords from the words that you already used for the title.

Abstract: please provide more detailed results about what you did find in this research. Some values should be shared in the abstract.

Equations: please use word equation toolbox instead of pasting as a picture.

Fig. 1: it looks like so amateur. Please redraw.

What is the repetition value of the analysis? The statistical part is so weak.

Principal component analysis (PCA) is used for decreasing the number of parameters that affects the process. So, how did you use PCA? It is not clear please give more details.

Tables: The article is overwhelmed with numerous tables, which decreases the reading motivation of the readers. Please transfer some tables to figures.

Optimization procedure can be done with experimental designs but here in this article I could not see any experimental design. At this point how could you explain the reliability of your optimization process?

Comments on the Quality of English Language

Minor editing of English language required.

Reviewer 2 Report

Comments and Suggestions for Authors

After reviewing the manuscript titled "Quality Optimization and Evaluation of New Cookie Product with Celery Root Powder Addition", I find it well written and the experiments were carefully planned and conducted.

Below are my detailed comments after reading the manuscript.

Point 2.2. In the freeze-drying process parameters only the condenser temperature is given, what was the shelf temperature and how was it changed?

Point 2.3. and 2.4. is: solem, ZBPP, Bydgszcz, Poland (error in the name of the town); instead it should be: Solem, zbpp, Bydgoszcz, Poland

Point 3.2. “With a percentage increase in celery powder content (2-7), there was an increase in cookies hardness, ranging from 2091 to 5299 N, for the cookie samples 1 and 7, respectively.” The increase in the hardness of the cookies is quite significant. Are there any indicated ranges of values ​​or data on this matter, above which hardness acceptability by consumers may be limited?

Point 3.3. and 3.4. In the discussion, it is indicated in several places that molasses residues affect the product (cookies) properties, such as color, but also taste and smell. What, although estimated, amount of molasses in relation to celery root could have been introduced with it into the final product as a result of choosing this method of removing water.

Table 4. In the explanation of abbreviations below the table, the letters a and b should probably be capitalized (A and B).

Tables 1, 6, 7, 8, 10. "% d.m." It is not explained anywhere what this unit is? What does the abbreviation % d.m. mean - it is not given in the text either.

Comments on the Quality of English Language

Minor editing of English language required

Reviewer 3 Report

Comments and Suggestions for Authors

In thids manuscript, the authors defined the optimal substitution amount of wheat flour with osmodehydrated and lyophilized celery root powder in the cookie, in which the most favourable technological and nutritive profiles were achieved in the first phase, In the second phase, the influence of different dehydration methods (combined osmodehydration and lyophilization, convective drying and lyophilization) on the same technological and nutritional parameters of cookies with the optimal addition of celery powder was examined. The authors compared the effect of celery root powder with different drying method on the quality of cookies, however, only 20% addition was used to compare, I don’t think it is enough.

In my oponin, the authors should investigate the drying method on the quality of celery root powder, including energy consumption, drying time, color, minerals, phenol content and antioxidant activity firstly. Then, the authors investigate the effect of the addition of celery root powdwer with different drying method on cookies quality at different ratios.

 Editing errors:

Please change “hours” to “h”;

Changge “the following equation” to “equation 1 or equation 2”

Round 2

Reviewer 1 Report

Comments and Suggestions for Authors

From my side, the article can be published as is.

Comments on the Quality of English Language

Minor editing of English language required.

Reviewer 3 Report

Comments and Suggestions for Authors

Accepted as it is